# Optimization of the Extraction Procedure for the Phenolic-Rich *Glechoma hederacea* L. Herb and Evaluation of Its Cytotoxic and Antioxidant Potential

**DOI:** 10.3390/plants11172217

**Published:** 2022-08-26

**Authors:** Karolina Grabowska, Kinga Amanowicz, Paweł Paśko, Irma Podolak, Agnieszka Galanty

**Affiliations:** 1Department of Pharmacognosy, Faculty of Pharmacy, Jagiellonian University, Medical College, Medyczna 9, 30-688 Kraków, Poland; 2Department of Food Chemistry and Nutrition, Faculty of Pharmacy, Jagiellonian University, Medical College, Medyczna 9, 30-688 Kraków, Poland

**Keywords:** *Glechoma hederacea*, extraction optimization, HPLC, quantification, phenolic acid, flavonoid, cytotoxicity, antioxidant activity

## Abstract

The dried *Glechoma hederacea* L. herb has a long history of use in traditional medicine. Its therapeutic potential is related to the presence of phenolic compounds. To optimize extraction efficiency the effect of the use of different techniques (HRE—heat reflux extraction, I/ME—infusion combined with maceration, UE—sonication and SE—Soxhlet extraction), various solvents (water and ethanol) and processing time (15 min to 2 h) on phenolics content was investigated. The HPLC method was applied to determine and compare the content of phenolic acids (rosmarinic, chlorogenic, protocatechuic) and flavonoids (rutin, isoquercetin) in the extracts. Furthermore, the cytotoxic activity of the extracts was examined for the first time against human cancer and normal cells of skin origin (A375, HTB140, HaCaT) and gastrointestinal origin (Caco-2 and HT-29, HepG2). In addition, the antioxidant potential was evaluated using the DPPH and FRAP method. The I/ME-water and HRE/ethanol procedures turned out to be optimal for obtaining extracts of dried *G. hederacea* L. herb rich in bioactive phenolics. These extracts exhibited high antioxidant activity, correlated with the content of the compounds analyzed. Furthermore, the extracts of the dried *Glechoma* herb were not toxic to normal human cells, indicating its safe use both internally and externally.

## 1. Introduction

Plants of the genus *Glechoma* (*Lamiaceae* family) are herbal products with a long history of use in Asian medicine, as well as in the western health system [1]. For example, *G. longituba* (Nakai) Kuprian is valued in Traditional Chinese Medicine as an effective agent against influenza, chronic pneumonia, haematuria, abnormal menstruation, various uterine diseases, leucorrhea, epilepsy, and rheumatoid arthritis [1,2]. In turn, *G. hederacea* L., commonly called ground ivy, which is the subject of our study, has been traditionally used both in Asia and Europe to treat gastritis, bronchitis, tinnitus, diarrhoea, cholelithiasis, and liver diseases [1,3,4,5]. Extracts of this plant species are also applied externally in various skin diseases [4,6]. The dried herb of *G. hederacea* L. is valued not only for its therapeutic properties but is also a commercially available food product, the most well-known example is gill tea, used in England since the 18th century [6]. Moreover, some breweries still use ground ivy as a bitter source in beer production [7], and dried aerial parts of the plant are gaining more and more popularity as a spice.

Recent studies on the biological activity of ground ivy have demonstrated its anti-inflammatory, antimelanogenic, antimicrobial, antimutagenic, antioxidant, antiacetylcholinesterase and neuroprotective activity [5,8,9,10,11,12,13,14,15]. 

Pharmacological effects are most often related to the presence of phenolics, such as phenolic acids and flavonoids, which are found in significant amounts in this plant species [12,16,17,18,19]. Nevertheless, phytochemical studies on *G. hederacea* L. aerial parts revealed also the presence of triterpenes and diterpenes, essential oil, alkaloids, fatty acids, and tannins [1,3,20,21,22,23,24]. 

Some recently published reports suggest that polyphenols from *G. hederacea* L. may have cytotoxic potential [14,16]. In one study, water extracts of dried *G. hederacea* L. were examined in the human laryngeal carcinoma cell line (HEp2) [16], while the second report focused on the activity of extracts and fractions obtained from fresh herb [14]. The activity of extracts prepared from dried plant material is especially interesting, due to the fact that ground ivy is used in traditional medicine and as a food product mainly as a dried herb in the form of water (e.g., infusion, decoction) and ethanol extracts [4,6,25]. 

Considering that phenolic compounds are presumed to be the main bioactive components of plant material it is essential to achieve the most effective procedure for their extraction. Although many reports provide data on the quantification of phenolics in *G. hederacea* L. [12,16,24,26,27,28,29], only few have focused on the influence of various extraction methods on the content of individual compounds [17,18]. Furthermore, data on the optimization of the extraction procedure of dried *G. hederacea* L. herb are limited to only one study, comparing different water extracts in terms of their antioxidant activity and total phenolic content (TPC) [18]. 

Therefore, the objective of the present study was to comparatively quantify the content of predominant phenolic acids and flavonoids in water and ethanol extracts prepared from dried *G. hederacea* L. herb with the use of different extraction protocols (heat reflux extraction, infusion combined with maceration, ultrasound-assisted extraction and extraction in the Soxhlet apparatus) and various processing times, in order to preselect the optimal procedure. 

Furthermore, data on the wide traditional use, both internally and externally, of dried *G. hederacea* L. herb and its potential cytotoxic activity prompted us to evaluate the efficacy and safety of the obtained extracts for human cancer and normal cell lines. The experiment was carried out on cells grouped in the skin panel (melanoma A375, HTB140, normal keratinocytes HaCaT) and in the gastrointestinal panel (colon adenocarcinoma Caco-2 and HT-29, hepatocellular carcinoma HepG2). Finally, the antioxidant potential of water and ethanol extracts from dried ground ivy herb, as a potential chemopreventive agent, was also evaluated and compared.

## 2. Results

### 2.1. HPLC Analysis of Water and Ethanol Extracts of Dried G. hederacea L. Herb Prepared by Various Extraction Procedures

The predominant compounds of the extracts examined were selected by qualitative HPLC analysis, and included three phenolic acids: rosmarinic (RA), chlorogenic (ChA) and protocatechuic (PCaA) acids and two flavonoids: rutin (Ru) and isoquercetin (IsoQ). The results of their quantification are shown in Table 1 and Table 2. The analysis indicates that the use of different solvents and extraction techniques affects the results of quantitative determination. 

Among phenolic acids, rosmarinic acid (RA) was the main one, with its highest content (4.28–4.89 mg/g dry plant material) in water extracts prepared by the I/ME method. The level of RA was significantly lower in ethanol extracts prepared by the same method (1.07–1.14 mg/g dry plant material) (Figure 1a). In the case of ethanol, extracts most abundant in RA were obtained using the HRE method (2.98–3.45 mg/g dry plant material), however, compared to the I/ME method with the use of water as extractant, the content of RA was significantly lower (Figure 1a). The other extraction procedures analyzed were characterized by a low efficiency of RA extraction, regardless of the solvent used. It can be seen that in the water extracts, the RA content decreased with the prolonged extraction time, but this difference was not statistically significant. This trend was not observed in ethanol extracts prepared with the HRE or I/ME method. In turn, extending the extraction time from 15 to 30 min in the case of ethanol extracts prepared with the UE method significantly increased the efficiency of the process. However, the use of the latter method and water as the extractant drastically decreased the level of RA to almost no detectable.

Chlorogenic (ChA) and protocatechuic (PCaA) acids were present in each of the extract samples. Similarly to RA, the highest levels of ChA were found in the water extracts prepared by the I/ME method (3.41–3.70 mg/g of dry plant material). As can be seen in Figure 1b, a lower ChA content was observed in both ethanol and water extracts prepared using the HRE method (1.02–1.32 and 1.52–2.04 mg/g of dry plant material), respectively. The other extraction procedures were characterized by a lower efficiency of ChA extraction, regardless of the solvent used, compared to the HRE method. The ChA content decreased with the prolonged extraction time in the case of the HRE extraction procedure. 

The concentration of PCaA determined in the samples was in the range of 0.02–0.73 mg/g of dry plant material. The extraction procedures can be ranked according to the decreasing PCaA content as follows: I/ME/water > HRE/2 h/water > HRE/1 h/2 h/ethanol > HRE/30 min ≈ SE/2 h/water > UE/water > I/ME/ethanol ≈ SE/ethanol > UE/ethanol (Table 1 and Table 2). 

The graphic view of the composition of the phenolic acids in the tested extracts is presented in Figure 2. It can be easily seen that their highest content (8.26–9.13 mg/g of dry plant material) was found in the water extracts prepared by the I/ME procedure (Figure 2). The sum of phenolic acids was significantly higher in the water extracts compared to the ethanol extracts (0.72–1.41 mg/g dry plant material) prepared with the same method (I/ME). Furthermore, extracts prepared using the HRE technique showed a lower content of phenolic acids sum compared to those obtained by the I/ME/water procedure (4.47–5.08 mg/g dry plant material). The differences in phenolic acid sum between ethanol and water extractions using HRE techniques were not significant. It can also be seen that the sum of phenolic acids in the extracts obtained with the use of the UE technique is low, regardless of the solvent used (0.25–1.40 mg/g of dry plant material).

The highest content of Ru and IsoQ was found in ethanol extracts prepared by the HRE method (0.84–0.99 and 0.82–0.96 mg/g of dry plant material, respectively). In the case of Ru, it was twice as high as that found in aqueous extracts prepared with the same extraction technique (Figure 3a). On the other hand, the I/ME procedure with the use of water allowed for a much more effective extraction of Ru compared to the use of ethanol as the extractant. However, the level of Ru in the water extracts prepared by I/ME was lower than in the ethanol extracts obtained by HRE (Table 1 and Table 2). The duration of the process did not significantly affect the efficiency of Ru extraction, regardless of the technique used. In turn, IsoQ content in the water extracts prepared with the HRE method increased significantly along with the extension of the extraction time. The efficiency of the two-hour HRE extraction with water was comparable to the use of a 30-min or two-hour HRE process with ethanol as extractant (Figure 3b).

### 2.2. Antioxidant Activity

To determine the antioxidant potential, two methods were used: FRAP and DPPH, and the results are shown in Table 3. The water extracts showed significantly higher antioxidant activity compared to the ethanol extracts prepared by the same method. This trend was seen in almost all extracts in both FRAP and DPPH, with the exception of extracts obtained by UE/1 h (in the DPPH test).

Generally, extracts prepared by I/ME and HRE method, regardless of the extraction time, revealed the highest antioxidant activity. The water extracts obtained by the I/ME method showed the highest activity, whereas the most active ethanol extract was prepared by the HRE technique. However, both the water and ethanol extracts obtained by these methods (I/ME and HRE) can be ranked according to the decreasing antioxidant activity (in both FRAP and DPPH tests) as follows: I/ME-water > HRE-water > HRE-ethanol > I/ME-water. 

The water extracts prepared by the other procedures (UE, SE) revealed significantly lower antioxidant activity in both tests compared to extracts obtained by the HRE and I/ME method. In turn, no significant differences were observed between the ethanol extracts prepared by SE and I/ME (30 min/1 h/2 h) and I/ME and UE (30 min/1 h). 

### 2.3. Cytotoxicity

The impact of the obtained extracts on human cancer and normal cells, representing the gastrointestinal tract and skin, was determined by MTT assay. Despite the wide range of the concentrations tested (10–100 µg/mL), only weak cytotoxicity was observed and for most samples the effect was not greater than untreated control cells (<10%). No significant differences were observed between the water and ethanol extracts prepared by different methods. Similarly, no differences in the response of particular cell lines, differing in their metastatic potential, were observed. What is more important, the extracts examined were characterized by high safety for normal skin cells used in the study, as they also did not reveal in vitro hepatotoxicity against HepG2 cells. 

## 3. Discussion

Plants abundant in phenolics are considered as health benefit products [30,31,32,33,34,35,36,37]. Several papers suggest that the pharmacological activity of *G. hederacea* L. has been linked to phenolic compounds [12,13,14,15,16]. Due to the fact that ground ivy is recommended not only in medicine but also in food industry in the form of extracts [4,6,25], the efficiency of extracting phenolics from the *Glechoma* herb is a key factor in obtaining bioactive extracts, rich in these compounds. Therefore, in the current study, a series of experiments was conducted to achieve the most effective procedure for phenolic extraction from dried ground ivy herb. 

Until now, in most studies aimed at quantitative analysis of phenolics in *G. hederacea* L., various extraction methods were used, but no direct comparison of the content of individual extract components was performed [12,16,24,26,27,28,29]. The only exception are two studies, which focused on water extracts obtained with the use of classical (HAE-heat assisted extraction) and modern (MAE-microwave-assisted and SWE-subcritical water extraction) methods [17,18]. In view of these incomplete and limited data, the conventional extraction methods that are still most commonly used in laboratories were compared.

In addition, to obtain better insight into the effectiveness of the extraction procedure, the analysis was focused not only on the extraction technique itself, but also on the type of extractant and processing time. 

In most of the published papers, the quantification of phenolics in *G. hederacea* L. was preceded by extraction with water [16,17,18,29,38], while other solvents, such as 70–80% MeOH [28,38] or 70% EtOH [24,38] or mixtures of solvents (e.g., methanol/acetone/water) were rarely used [27]. Only one study compared water and alcoholic (70%) extracts of *G. hederacea* L. obtained with the use of maceration (24 h; room temperature) [38]. 

Therefore, two extractants that are the most frequently used in pharmacy, food industry, and at home, namely water and ethanol, were chosen for the experiment. To compare the efficiency of the methods, the same amount of extractant was used in all procedures.

In the first step of the study, a direct qualitative and quantitative comparison of the individual phenolic compounds was performed in the obtained extracts. The qualitative profile was consistent with the results of the other authors [12,17,18,24,29]. However, isoflavones, such as daidzin, genistein, and genistin, which have been reported by Chou et al. [12] were not detected in any sample, and this result is consistent with the observations of Šeremet et al. [18]. In addition, in both water and ethanol extracts, protocatechuic acid and isoquercetin were identified. This is the first information on these phenolic compounds in this plant species.

The results of the quantitative study revealed that rosmarinic acid and chlorogenic acid are predominant components in extracts of the dry herb of *G. hederacea* L. Among flavonoids, rutin was detected as the main compound. These observations are consistent with other studies on *G. hederacea* L. [17,18,24,27,29]. However, published reports have indicated large differences in the content of RA, ChA and Ru in *G. hederacea* L. extracts (0.12–24.1, 0.02–8.4, 0–8.83 mg/g of dry plant material, respectively) [16,17,18,24,27,28,29,38]. These differences may be the result of variation due to environmental factors and the time of harvest [24,29,39]. Moreover, they can also result from differences in the details of the extraction procedures, so it is impossible to unequivocally compare the data between the experiments. 

In addition to RA, ChA, and Ru, protocatechuic acid (PCaA) and isoquercetin (IsoQ) were also quantified in the current investigation. To the best of our knowledge, this study is the first report on the quantitative determination of these compounds in *G. hederacea* L.

The highest extraction efficiency (mg of substance/dry plant material) of predominant phenolic acids (RA and ChA) was obtained for the I/ME procedure with water as an extractant. In turn, the use of the same extraction procedure with ethanol resulted in a low content of all the compounds analyzed in extracts of the *G. hederacea* L. herb. Similar observations of differences in ChA content in alcoholic and water macerations of *G. hederacea* L. (24 h; room temperature) were made by Oalđe et al. [38]. 

The results of our study indicate that in the case of ethanol as the extractant solvent, the HRE method is much more suitable for obtaining extracts rich in polyphenols from the *Glechoma* herb than the I/ME technique, ultrasound assisted extraction (UE) or Soxhlet extraction. 

In the current study, extracts obtained by HRE/ethanol had the highest content of flavonoids. This is probably due to the fact that the tested flavonoids: Ru and IsoQ have higher solubility in ethanol than in water [40,41,42] and their solubility increases with increasing solvent temperature [42], which may explain the efficiency of the HRE method. It can also be seen that, in the case of ethanol, the use of a temperature-assisted extraction technique, such as HRE, increases the efficiency of the process for dominant phenolic acids (RA and ChA) as compared to the other techniques. This observation is consistent with the results of Jacotet-Navarro et al. [43], in which a higher content of RA was observed in ethanol/water extracts (90:10 v/v) from *Rosmarinus officinalis* L. leaves prepared by HRE compared to the ME or UE method. However, it should be mentioned that UE is a method recommended by some authors in extracting RA from plant material [44,45,46], but the efficiency of the sonication process depends on many factors, such as frequency, temperature, and type of solvent [46,47]. Tungmunnithum et al. [46] revealed that sonication with the use of ethanol is much more effective in RA extraction from *Plectranthus scutellarioides* (L.) R.Br. leaves compared to UE with the use of water. Moreover, the extension of the duration of sonication with the use of ethanol increased the concentration of RA in extracts [46]. In our study we observed a similar tendency.

Although lower extraction efficiency of compounds was found using the UE/ethanol technique compared to the HRE/ethanol technique, it should be noted that the efficiency of the 30 min sonication was comparable to the four-time longer extraction of the plant material with ethanol in the Soxhlet apparatus.

Taking into account the quantitative results obtained for the two-hour extraction in the Soxhlet apparatus and HRE, it can be noticed that the higher efficiency of extraction of both phenolic acids and flavonoids was observed for HRE, which may indicate that immersion of the plant material directly in the boiling solvent increased the efficiency of the process. 

A similar trend in the effectiveness of extraction for dominant phenolic acids can be seen when comparing the results of HRE water extraction with SE or UE procedures. In our study, the use of ultrasonically assisted extraction at room temperature (25 °C) did not provide a sufficient yield in the extraction of compounds. 

Nevertheless, it should be noted that the use of the heat-assisted technique (HRE) with water as an extractant is a procedure that seems to have some limitations. Although the total amount of phenolic acids was comparable in the water and ethanol extracts obtained by the HRE technique, the results of the analysis showed that the content of some phenolics in the water extracts decreased with the extension of the extraction time. This effect was significant for ChA and noticeable, but statistically insignificant for the RA content. In the case of simple phenolic acids, such as PCaA, the effect was seen when the extraction time was extended from 1 to 2 h. In the case of HRE extraction with ethanol, this effect was noticeable only in the case of ChA. This reduction in the content of phenolic acids, especially depsides, may be due to their thermal degradation in water and alcoholic solutions [48,49,50]. 

The results of our research indicate that the time in which the herb of *G. hederacea* L. is exposed to boiling water may be a key factor in obtaining extracts rich in phenolics. In the I/ME procedure, the plant material is immersed in the solvent all the time, but the exposure time to the boiling extractant is short because the material is only poured over with boiling water, the temperature of which is lowered during the maceration process. The results of the qualitative study indicate that this procedure proved to be the optimal technique for obtaining aqueous extracts of *G. hederacea* L. This observation is especially interesting because the I/ME procedure is widely used during the preparation of ground ivy herb extracts by patients at home.

As several studies suggest the cytotoxic potential of polyphenols and extracts rich in polyphenolics [51,52,53,54], and polyphenols are generally characterized by high antioxidant properties, biological activity of the examined extracts of *G. hederacea* L., directed at the cytotoxic and antioxidant potential, was further performed. This was to verify the role of the dried plant in supporting and/or preventing some diseases, including gastritis, liver, or skin problems, reported in traditional medicine. Thus, for the cytotoxic study we chose two colon cancer (Caco2, HT29) and two melanoma cell lines (HTB140, A375). To determine the selectivity and safety of the extracts, hepatoma cells HepG2, revealing the phenotype of normal hepatocytes, and normal skin keratinocytes (HaCaT) were also included. This was the first attempt to examine the impact of *G. hederacea* L. extracts on such a wide panel of cancer and normal cells. Until now, the cytotoxic activity of the extracts from the dried plant has been described only once, in human laryngeal carcinoma Hep2 cells, and the effect was moderate [16]. Hot water extract from the fresh *Glechoma hederacea* L. plant was tested against HepG2 cells at concentrations as high as 400 µg/mL and a decrease of approximately 50% in cell viability was observed at 100 µg/mL, while in our study at this concentration the cell viability treated with water extracts decreased only slightly (approximately 10%) [14]. Such a vast difference may result from the different chemical compositions of the fresh and dried *Glechoma* plant, but also from the details in the preparation of the extract. Furthermore, our study did not demonstrate any toxic effect of the extracts tested in normal HaCaT, and HepG2 cells, supporting the traditional internal and external use of the plant in terms of its safety. Although further studies are undoubtedly needed, our preliminary results can be a recommendation to use the dried *Glechoma* plant as tea or spice.

Referring to the traditional and ethnobotanical use of *Glechoma hederacea* L., chemopreventive benefit of the plant to the organism was also examined, in terms of its possible protection from the excess of free radicals, which usually accompanies various diseases. Therefore, in the last step of our experiment, the antioxidant potential of the extracts was evaluated and compared. The measured antioxidant capacity of *G. hederacea* L. water extracts by the DPPH assay showed a good correlation with PCaA (r = 0.98, *p* < 0.05), RA (r = 0.95, *p* < 0.05) and ChA (r = 0.92, *p* < 0.05) while those of the FRAP assay were not significant. In the case of ethanolic extracts, significant good correlations were observed for all phenolic acids evaluated with both antioxidant methods used (FRAP and DPPH). For rutin and isoquercetin, the significant correlation with DPPH was very good (r = 0.94, r = 0.74, respectively), but the correlation coefficient with FRAP was lower (r = 0.85, r = 0.66, respectively), but still significant. Belščak-Cvitanović et al. [16] observed the same effect as in our study, *Glechoma* extracts prepared by infusion had the highest antioxidant activity measured by the FRAP method compared to their macerated and decocted extracts. Gwiazdowska et al. [10] noted the highest Pearson correlation coefficient between FRAP and total phenolic compounds followed by DPPH; this observation is in opposition to our results for FRAP. These authors suggested that the total phenolics in the water extracts of *G. hederacea* L. were the primary contributor to antioxidant capacity. In the case of our study, the lack of this phenomenon should be evaluated in a further study on other water-soluble compounds with antioxidant activity. Especially Matkowski et al. [28] noted that nonpolar fractions of *Glechoma* were practically inactive in radical scavenging. 

Interesting results were published by Oalđe et al. [38] who found that methanolic and ethanolic extracts of *Glechoma* showed a higher DPPH inhibition activity compared to water extracts, and exhibited the lowest DPPH radical inhibition. The results showed that the ethanolic extract of *Glechoma* was the only one with DPPH scavenging capacity similar to the positive control, 2-*tert*-butyl-4-hydroxyanisol.

Chou et al. [13] proposed the antioxidant mechanism for the water extract of *Glechoma hederacea* L. It exerts an antioxidant effect on the cycling of oxidized materials (Ox’) by reducing oxidants and/or avoiding lipid oxidation by chelating metal ions (Fe^2+^). Additionally, the hot water extract of *G. hederacea* L. was safe in terms of the antioxidant activity [13]. This observation was in opposition to the results obtained by Gwiazdowska et al. [10] who noted that the differences between the antioxidant activity of the extract prepared at 40–60 °C differed significantly in the case of DPPH, FRAP, and ABTS.

## 4. Materials and Methods

### 4.1. Standards and Reagents

HPLC grade acetonitrile, methanol and formic acid were purchased from Sigma-Aldrich (St. Louis, MO, USA). Standards for HPLC analysis (phenolic acids: rosmarinic acid, chlorogenic acid, and protocatechuic acid and flavonoids: rutin and isoquercetin) were purchased from Fluka Chemie. All reagents were of analytical grade. Distilled water and ethanol were purchased from Sigma-Aldrich. Trolox (6-hydroxy-2,5,7,8,-tetramethyl-chroman-2-carboxylic acid); and FeCl_3_·6H_2_O; 1,1-diphenyl-2-picrylhydrazyl (DPPH) were from Sigma Aldrich. 2,4,6-Trispyridyl-s-triazine (TPTZ) was purchased from Fluka Chemie (Buchs, Switzerland). Ethanol, acetic acid, ammonium hydroxide solution, hydrochloric acid, sodium acetate, and sodium carbonate were purchased from Avantor Performance Materials Poland S.A. (Gliwice, Poland). Dulbecco’s Modified Eagle’s Medium High Glucose (4500 mg/L), Dulbecco’s Modified Eagle’s Medium F12 HAM, Triton X-100 (X100), penicillin-streptomycin solution, trypsin-EDTA solution, phosphate-buffered saline PBS, MTT reagent, DMSO were purchased from Sigma-Aldrich. 

### 4.2. Plant Material

*Glechoma hederacea* L. was collected during the reproductive (flowering) phase in May 2020 from the Botanical Garden, Medical College, Jagiellonian University, Kraków, Poland (50°00′44.3″ N 19°59′38.4″ E). The identity of the plant material was confirmed by Dr Agnieszka Szewczyk from the Department of Pharmaceutical Botany Jagiellonian University, Kraków, Poland: A voucher specimen (Reference No. FG/Ghed/2020/01) has been deposited at the Department of Pharmacognosy, Faculty of Pharmacy, Medical College, Jagiellonian University, Cracow, Poland. The aerial parts of *G. hederacea* L. were air-dried in the dark, at 24 °C in an air-conditioned room to a constant weight. The plant material was powdered and stored in airtight containers. 

### 4.3. Extraction Procedures

The powdered aerial parts of *G. hederacea* L. were accurately weighed (2.0 g) and then extracted with ethanol (EtOH) and water (H_2_O). Different techniques (heat reflux extraction—HRE; maceration—ME, sonication—UE, and Soxhlet extraction—SE) and various extraction times were used. Six extracts of plant material were prepared for each extraction method. Extractions were performed as follows:

Heat reflux extraction (HRE): The plant material was placed in a round bottom flask and extracted with 50 mL of solvent (ethanol or water) under cooler with the use of a heating mantle. The extraction procedure was carried out for 30 min, 1 h or 2 h.

Infusions/Maceration (I/ME): 2.0 g of plant material was placed in a flask. The plant material was then poured with 50 mL of an appropriate boiling solvent (ethanol or water) and extracted for 30 min, 1 h or 2 h at room temperature.

Ultrasonic extraction (UE): The dried plant material was placed in a conical flask and extracted with 50 mL of ethanol or water for 15 min, 30 min or 1 h at 30 °C. Ultrasound-assisted extraction was performed with the use of Sonic-3 ultrasonic bath, (POLSONIC, Warsaw, Poland).

Soxhlet extraction (SE): 2.0 g of plant material was placed in a cellulose thimble and extracted in Soxhlet apparatus with 50 mL of appropriate solvent for 2 h. 

### 4.4. Sample Preparation

The extracts obtained from *G. hederacea* L. were filtered using quantitative filter papers (POCH, Gliwice) and concentrated under reduced pressure on a rotary evaporator to obtain a volume of 25 mL. The extracts were then transferred to a 25 mL volumetric flask. 

Preparation of samples for HPLC analysis: 1.5 mL of each extract was filtered using Titan2 nylon HPLC filters (0.45 μm pore size) and transferred to vials. Samples were stored in a freezer (−20 °C) for further HPLC analysis.

Preparation of samples for cytotoxicity tests: 15 mL of each extract was evaporated to dryness under reduced pressure, using a rotary evaporator (50 °C) and weighed to a constant mass. Samples were stored in a freezer (−20 °C) for further analysis.

Preparation of samples for antioxidant activity tests: 5 mL of each extract was transferred to amber glass vials. Samples were stored in a freezer (−20 °C) for further analysis.

### 4.5. HPLC Analysis

For the determination of polyphenols in plant extract samples the HPLC method, previously described [55] was used. The HPLC system consisted of a Dionex 100 (Dionex Softron GmbH, Germering, Germany) with a Photodiode 100 detector. A volume of 20 μL of each extract was injected into Hypersil Gold (C-18) column (5 μm, 250 × 4.6 mm, Thermo Scientific, Runcorn, UK), and analyzed with the mobile phase: 1% formic acid (A) and acetonitrile (B) in a gradient mode 5–60% B for 60 min. For the identification of the obtained peaks, their retention times and UV spectrum were compared with those of the standards. The amount of phenolic acids: rosmarinic acid, chlorogenic acid, and protocatechuic acid, and flavonoids: rutin, and isoquercetin, was calculated using the appropriate standard curves prepared from the solutions of reference substances. All analyzes were performed in triplicate and the mean values were expressed as mg/1 g dry plant material.

### 4.6. Preparation of Standard Solutions and Calibration Curves

Each reference substance was weighed to 5 mg in a volumetric flask using an analytical balance and dissolved in 5 mL of methanol to make 1 mg/mL stock solution. Standard curves were prepared based on a series of dilutions in the range of 0.0625–1.0 mg/mL. Each dilution was analysed in triplicate with HPLC.

### 4.7. Cytotoxic Activity

The experiment was carried out on two sets of human cancer and normal cell lines, purchased from ATCC (American Type Culture Collection; Manassas, VA, USA): skin panel (melanoma HTB140, derived from metastatic site: lymph node, ATCC Hs 294T; malignant melanoma A375, ATCC CRL-1619; skin keratinocytes HaCaT) and gastrointestinal panel (colorectal adenocarcinomas Caco-2, ATCC HTB-37, HT-29, ATCC HTB- 38; hepatocellular carcinoma HepG2, ATCC HB-8065), and cultured under conditions described previously [56]. Cells were seeded in 96-well plates for 24 h (1.5 × 10^4^ cells/well), and then a fresh medium containing different concentrations of the extracts tested (10–100 μg/mL) was added. The incubation lasted 24 h. Cell viability was measured by MTT assay, as previously described [57]. The absorbance at 490 nm was measured using a Biotek Synergy microplate reader (BioTek Instruments Inc., Winooski, VT, USA). Cell viability was expressed as percent of dead cells. Each experiment was carried out in triplicate. Doxorubicin was used as a reference standard.

### 4.8. Determination of Antioxidant Activity

The analysis was performed using the DPPH and FRAP methods, as previously described [58]. Briefly, the DPPH methanolic solution (3.9 mL, 25 mg/L) was added to the extracts tested (0.1 mL) and the reaction was monitored at 515 nm until the absorbance was constant. Each sample was measured in three replicates. The mean capacity was expressed as µM Trolox/g dry plant material. A fresh working FRAP solution (900 μL; 2.5 mL 10 mM ferric-tripiridyltriazine in 40 mM HCl, 2.5 mL 20 mM FeCl_3_·H_2_O and 25 mL 0.3 mol/L acetate buffer, pH 3.6) was added to 90 μL of distilled water and 30 μL of the extracts tested, and the absorbance of the mixture was measured at 593 nm. All analyzes were performed in triplicate. The mean capacity was expressed as µM Fe^2+^/g dry plant material. The absorbance of both antioxidant assays was measured using a Biotek Synergy microplate reader (BioTek Instruments Inc., Winooski, VT, USA).

### 4.9. Statistical Analysis 

The quantitative data obtained were analyzed using Statistica v.13.3 (StatSoft). Each variable was expressed as the mean (±SD). The statistical significance between extracts in the quantification study was determined using analysis of variance (Welch’s ANOVA) and the post hoc Tukey multiple comparison test. One-way analysis of variance (ANOVA) and the post hoc Tukey multiple comparison test were used to check the differences between extracts in the antioxidant study. Differences between two extracts were tested using the *t*-Student test. Pearson’s correlation coefficients and *p*-values were used to show the correlations and their significance. The probability level of *p* < 0.05 was considered statistically significant.

## 5. Conclusions

The results of the current study indicate that not only the use of different extraction techniques but also the selection of the appropriate extractant for the method significantly affects the efficiency of the process and, consequently, the content of analyzed compounds in extracts. The highest extraction efficiency of predominant phenolics was obtained for water extracts prepared by the I/ME procedure, followed by the HRE method with ethanol as the extractant. The extracts obtained using these procedures were characterized by high antioxidant activity, which is correlated with the content of analyzed phenols. This may be useful in promoting dried *Glechoma* herb as a chemopreventive agent. Additionally, our research has shown that the extracts of dried *Glechoma* herb were not toxic to normal human liver and skin cells, indicating their safe use both internally and externally. Our research showed that the I/ME-water procedure turned out to be optimal for obtaining extracts from the dried herb of *G. hederacea* L. rich in bioactive phenolics. This extraction protocol is simple, convenient, and does not require special laboratory equipment. Therefore, our research justifies the idea of preparing phenolics-rich ground ivy water extracts by patients at home.

## Figures and Tables

**Figure 1 plants-11-02217-f001:**
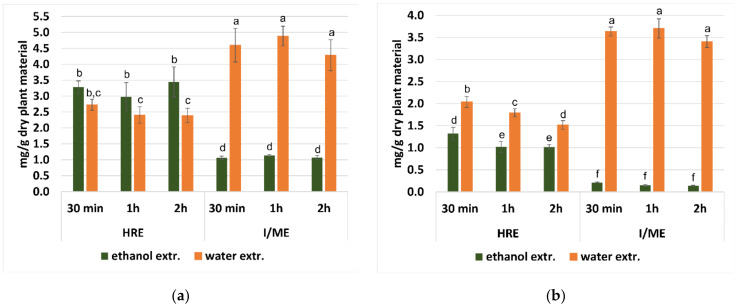
The content (mg/g dry plant material) of rosmarinic acid (RA) (**a**) and chlorogenic acid (ChA) (**b**) in water and ethanol extracts from dried aerial parts of *G. hederacea* L., prepared by pouring boiling solvent followed by maceration for 30 min, 1 h or 2 h at room temperature (I/ME) and heat reflux extraction (HRE). Results are presented as the mean ± SD (standard deviation) calculated from six independent experiments. The means with the same letter are not significantly different from each other (*p* > 0.05, F Welch’s ANOVA, followed by Tukey’s multiple comparison test), within the samples analyzed.

**Figure 2 plants-11-02217-f002:**
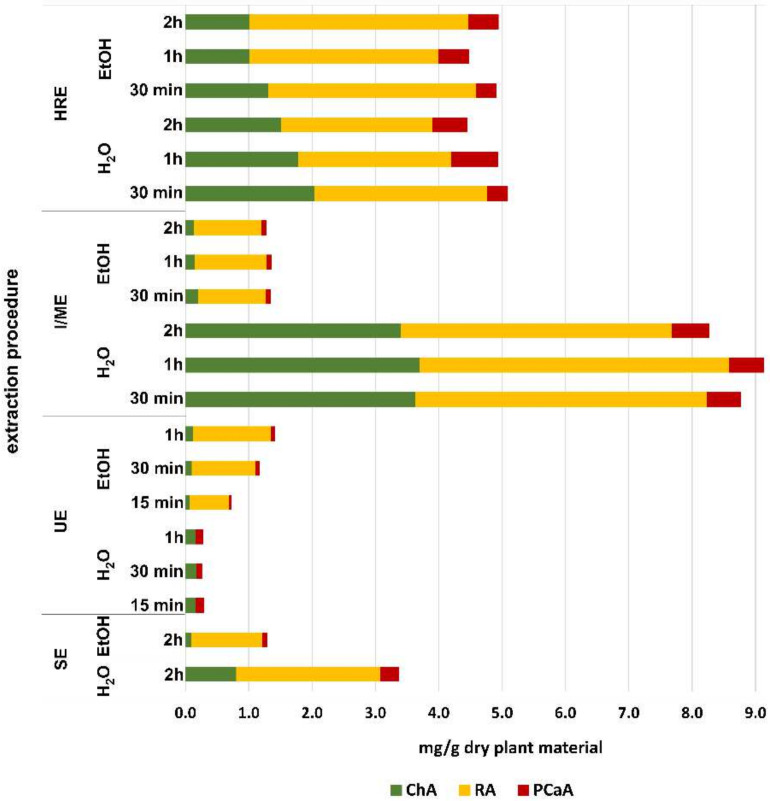
Sum (mg/g dry plant material) of determined phenolic acids: chlorogenic acid (ChA) and rosmarinic acid (RA) in water and ethanol extracts from dried aerial parts of *G. hederacea* L., prepared by different extraction methods. Abbreviations: HRE—heat reflux extraction; UE—ultrasound-assisted extraction; SE—extraction in the Soxhlet apparatus; I/ME—extracts prepared by pouring boiling solvent followed by maceration for 30 min, 1 h or 2 h at room temperature.

**Figure 3 plants-11-02217-f003:**
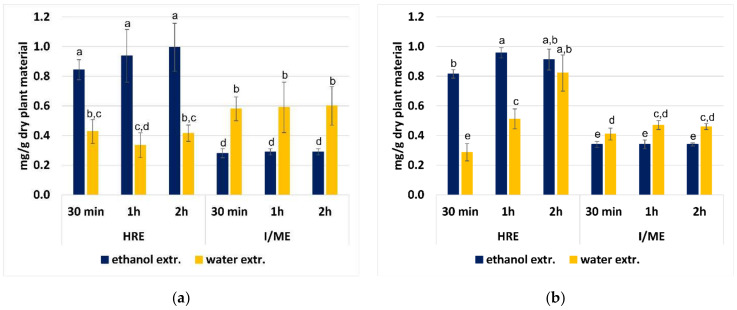
The content (mg/g dry plant material) of rutin (Ru) (**a**) Description of what is contained in the first panel; (**b**) Description of what is contained in the second panel. Figures should be placed in the main text near to the first time they are cited. A caption on a single line should be centered. and isoquercetin (IsoQ) in water and ethanol extracts from dried aerial parts of *G. hederacea* L., prepared by pouring boiling solvent followed by maceration for 30 min, 1 h or 2 h at room temperature (I/ME) and heat reflux extraction (HRE). Results are presented as the mean ± SD (standard deviation) calculated from six independent experiments. The means with the same letter are not significantly different from each other (*p* > 0.05, F Welch’s ANOVA, followed by Tukey’s multiple comparison test), within the samples analysed.

**Table 1 plants-11-02217-t001:** Average content of phenolic acids (chlorogenic acid, rosmarinic acid, protocatechuic acid) and flavonoids (rutin and isoquercetin) in ethanol extracts from dried aerial parts of G. hederacea L., prepared with the use of different extraction procedures.

Extraction Procedure		Mean Content of Compound ± SD [mg/g Dry Plant Material] in Ethanol Extracts from Dried Herb of *G. hederacea* L.				
Method	Time	Chlorogenic Acid	Rosmarinic Acid	Protocatechuic Acid	Rutin	Isoquercetin
HRE	30 min	1.32 ± 0.14 ^a^	3.28 ± 0.20 ^a^	0.30 ± 0.03 ^a^	0.84 ± 0.07 ^a^	0.82 ± 0.06 ^a^
	1 h	1.02 ± 0.13 ^b^	2.98 ± 0.44 ^a^	0.47 ± 0.05 ^b^	0.94 ± 0.18 ^a^	0.96 ± 0.09 ^b^
	2 h	1.02 ± 0.06 ^b^	3.45 ± 0.67 ^a^	0.47 ± 0.02 ^b^	0.99 ± 0.16 ^a^	0.91 ± 0.12 ^b^
I/ME	30 min	0.21 ± 0.01 ^c^	1.07 ± 0.05 ^b,c^	0.06 ± 0.002 ^c^	0.28 ± 0.03 ^b,c^	0.34 ± 0.02 ^c^
	1 h	0.15 ± 0.02 ^c,d^	1.14 ± 0.03 ^b,c^	0.06 ± 0.003 ^c^	0.29 ± 0.02 ^b,c^	0.34 ± 0.03 ^c,d^
	2 h	0.14 ± 0.01 ^c,d^	1.07 ± 0.06 ^b,c^	0.06 ± 0.002 ^c^	0.29 ± 0.02 ^b,c^	0.34 ± 0.01 ^c,d^
UE	15 min	0.07 ± 0.01 ^d^	0.62 ± 0.03 ^c^	0.02 ± 0.004 ^d^	0.18 ± 0.01 ^c^	0.24 ± 0.02 ^d^
	30 min	0.11 ± 0.01 ^c,d^	1.01 ± 0.12 ^b,c^	0.05 ± 0.002 ^c,d^	0.25 ± 0.02 ^b,c^	0.37 ± 0.04 ^c^
	1 h	0.12 ± 0.01 ^c,d^	1.22 ± 0.16 ^b^	0.06 ± 0.01 ^c^	0.30 ± 0.05 ^b^	0.38 ± 0.04 ^c^
SE	2 h	0.10 ± 0.01 ^c,d^	1.12 ± 0.06 ^b,c^	0.06 ± 0.002 ^c^	0.33 ± 0.04 ^b^	0.34 ± 0.02 ^c,d^

Results are presented as the mean ± standard deviation (SD) calculated from six independent experiments. Abbreviations: HRE—heat reflux extraction; I/ME—infusion/maceration, UE—ultrasonic extraction; SE—Soxhlet extraction; Results marked with the same letter within each column did not differ significantly (*p* > 0.05).

**Table 2 plants-11-02217-t002:** Average content of phenolic acids (chlorogenic acid, rosmarinic acid, protocatechuic acid) and flavonoids (rutin and isoquercetin) in water extracts of dried aerial parts of *G**. hederacea* L., prepared with the use of different extraction procedures.

Extraction Procedure		Mean Content of Compound ± SD [mg/g Dry Plant Material] in Water Extracts from Dried Herb of *G. hederacea* L.				
Method	Time	Chlorogenic Acid	Rosmarinic Acid	Protocatechuic Acid	Rutin	Isoquercetin
HRE	30 min	2.04 ± 0.12 ^a^	2.73 ± 0.17 ^a^	0.31 ± 0.02 ^a^	0.43 ± 0.08 ^a^	0.29 ± 0.03 ^a^
	1 h	1.79 ± 0.09 ^b^	2.41 ± 0.26 ^a^	0.73 ± 0.09 ^b^	0.34 ± 0.08 ^a,c^	0.51 ± 0.03 ^b^
	2 h	1.52 ± 0.10 ^c^	2.39 ± 0.23 ^a^	0.54 ± 0.05 ^c^	0.42 ± 0.06 ^a^	0.82 ± 0.07 ^c^
I/ME	30 min	3.64 ± 0.10 ^d^	4.60 ± 0.53 ^b,c^	0.52 ± 0.02 ^c^	0.58 ± 0.08 ^b^	0.41 ± 0.04 ^d^
	1 h	3.70 ± 0.22 ^d^	4.89 ± 0.31 ^b^	0.54 ± 0.02 ^c^	0.59 ± 0.19 ^b^	0.47 ± 0.03 ^b,d^
	2 h	3.41 ± 0.13 ^d^	4.28 ± 0.49 ^c^	0.57 ± 0.04 ^c^	0.60 ± 0.13 ^b^	0.46 ± 0.02 ^b,d^
UE	15 min	0.17 ± 0.03 ^e^	tr	0.11 ± 0.01 ^d^	tr	tr
	30 min	0.19 ± 0.04 ^e^	tr	0.07 ± 0.02 ^d^	tr	0.02 ± 0.03 ^e^
	1 h	0.17 ± 0.01 ^e^	tr	0.10 ± 0.02 ^d^	tr	0.04 ± 0.01 ^e^
SE	2 h	0.81 ± 0.07 ^f^	2.27 ± 0.07 ^a^	0.29 ± 0.02 ^a^	0.21 ± 0.07 ^c^	0.18 ± 0.07 ^f^

Results are presented as the mean ± standard deviation (SD) calculated from six independent experiments. Abbreviations: HRE—heat reflux extraction; I/ME—infusion/ followed by maceration, UE—ultrasonic extraction; SE—Soxhlet extraction; tr—traces; Results marked with the same letter within each column did not differ significantly (*p* > 0.05).

**Table 3 plants-11-02217-t003:** Antioxidant activity of extracts from dried aerial parts of *G. hederacea* L., prepared with the use of different extraction procedures.

Extraction Procedure		FRAP [µM Fe^2+^/g Dry Plant Material]		DPPH [µM Trolox/g Dry Plant Material]	
Method	Time	Water Extracts	Ethanol Extracts	Water Extracts	Ethanol Extracts
HRE	30 min	238.71 ± 18.97 ^a^	155.03 ± 8.52 ^a^	133.75 ± 15.39 ^a^	72.95 ± 2.99 ^a^
	1 h	233.49 ± 4.62 ^a^	162.83 ± 15.01 ^a^	123.92 ± 4.23 ^a^	94.61 ± 11.05 ^a^
	2 h	234.91 ± 14.87 ^a^	167.87 ± 9.10 ^a^	129.61 ± 1.94 ^a^	103.99 ± 11.83 ^a^
I/ME	30 min	301.26 ± 3.17 ^b^	41.17 ± 2.01 ^b^	156.15 ± 15.08 ^b^	23.00 ± 2.22 ^b^
	1 h	296.10 ± 2.03 ^b^	43.78 ± 2.34 ^b^	167.40 ± 4.72 ^b^	21.62 ± 0.75 ^b^
	2 h	295.27 ± 6.20 ^b^	42.64 ± 1.09 ^b^	162.76 ± 11.51 ^b^	21.54 ± 2.44 ^b^
UE	15 min	212.37 ± 3.88 ^c^	22.21 ± 1.79 ^b^	23.91 ± 1.66 ^c^	11.52 ± 1.09 ^b^
	30 min	67.41 ± 3.52 ^d^	35.14 ± 3.06 ^b^	25.39 ± 0.77 ^c^	17.49 ± 0.46 ^b^
	1 h	59.90 ± 3.51 ^d^	42.00 ± 7.30 ^b^	23.13 ± 4.26 ^c^#	21.83 ± 3.19 ^b^#
SE	2 h	148.73 ± 2.84 ^e^	42.43 ± 0.55 ^b^	67.68 ± 2.34 ^d^	20.47 ± 1.04 ^b^

The results are presented as the mean ± standard deviation (SD) calculated from six independent experiments. Abbreviations: HRE—heat reflux extraction; I/ME—infusion/maceration, UE—ultrasonic extraction; SE—Soxhlet extraction; Results marked with the same letter within each column did not differ significantly. The means with # are not significantly different from each other in each row within the test (FRAP or DPPH).

## Data Availability

Data is contained within the article.

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
