# Peer review of "Optimization of the Extraction Procedure for the Phenolic-Rich Glechoma hederacea L. Herb and Evaluation of Its Cytotoxic and Antioxidant Potential"

_plants, 2022, doi:10.3390/plants11172217_

Round 1

Reviewer 1 Report

The manuscript is well organized, results are clearly presented and references are up-to-date.

There are only some minor issues that should be addressed by authors.

Please rewrite Materials and Methods to decrease similarity with DOIs: 10.1055/a-1345-9377, 10.3390/ph15070806 and 10.1016/j.fbio.2021.100888.

Line 420: How did you concentrate the extracts?

Line 430: Please add the volume of the extract used in HPLC analysis.

Line 459: Please add hours after 24.

Table 3, Line 469: Please check if unit for DPPH is μM Trolox/g or μmol Trolox/g.

Lines 477-479: Please delete duplicate sentences.

Author Response

Dear Reviewer,

We are very grateful for the time you devoted to read our manuscript and for your helpful comments and suggestions. We did our best to make appropriate changes and correct the text so we hope our manuscript has improved. All suggested changes are highlighted in yellow.

Below please find detailed responses to all queries:

Comment: The manuscript is well organized, results are clearly presented and references are up-to-date. There are only some minor issues that should be addressed by authors. Please rewrite Materials and Methods to decrease similarity with DOIs: 10.1055/a-1345-9377, 10.3390/ph15070806 and 10.1016/j.fbio.2021.100888.

Response: The mentioned papers are in fact the manuscripts prepared and written by our research team, that is why some similar expressions occurred in the Material and Methods paragraphs, as we use these methods for many years. However, we did some efforts to rewrite this part of the manuscript to decrease the level of similarity.

Comment: Line 420: How did you concentrate the extracts?

Response: We concentrated the extracts under reduced pressure in a rotary evaporator. We have added this information in the Material and Methods section; 4.4. Sample preparation: "The extracts obtained from G. hederacea L. were filtered using quantitative filter papers (POCH, Gliwice) and concentrated under reduced pressure on a rotary evaporator to obtain a volume of 25 mL. "

Comment: Line 430: Please add the volume of the extract used in HPLC analysis.

Response: The information was added (see paragraph 4.5)

Comment: Line 459: Please add hours after 24.

Response: We added the word "hours" in the indicated place.

Comment: Table 3, Line 469: Please check if unit for DPPH is μM Trolox/g or μmol Trolox/g.

Response: The units in DPPH test should be μM Trolox/g. We checked that they are uniform throughout the manuscript.

Comment: Lines 477-479: Please delete duplicate sentences.

Response: The duplicate sentence has been removed.

Reviewer 2 Report

In this paper, selected phenolic acids and flavonoids in various extracts f Glechoma hederacea prepared by different extraction methods owere analyzed by HPLC. In addition, antioxidant potential and cytotoxicity of the extracts were also determined.

Giving the plant Latin binomial names, the genus and species names in Latin should be in italic, which must be accompanied by their authority names, which should not be italic. Please indicate author name in the title, too.

Actually, please check all plant names and their families according to APG4 plant classification system as well as IPNI or the Plant List. Use the accepted names and new family names to be the most correct.

Plant family Latin names are not written in italic. Please correct.

Grammatical and typological errors are present and should be rectified. For example;

line 17, “the cytotoxic activity of extracts” should be corrected as “cytotoxic activity of the extracts”.

line 30, author names for G. longituba, Rosmarinus officinalis, Plectranthus scutellarioides?

line 164, correct as “IsoQ content in the water extracts”.

line 204, correact as “was determined by MTT assay”.

On which bases, did you select these particular phenolic acids and flavonoids in HPLC analyses? Why not more?

"per", "via", "versus", "e.g.", "i.e.", “viz.”  italic in all cases.

“p” (statistical) italic in all cases.

Always use 3rd person style in the text. Not like “we decided to conduct …

line 332, please specify the plant species such as in “Hot water extract from the fresh Glechoma plant”

line 344, which ethnobotanical use of the plant did you correlate the mentioned effect in “….wanted to verify whether the use of the plant may have a chemopreventive benefit….”?

Please pay attention to punctuation marks.

line 388, correct as “phosphate-buffered

line 378, correct as “phenolic acids”.

post hoc should be italic in all cases.

Glechoma must be italic including references such as no. 1 and 10.

In “3rd”, rd must be superscripted.

In references, paper titles should be lower case except the first word in sentence.

line 579, correct as “Content of p-coumaric”, p should be italic in all cases.

Ref. 49, the terms "cis” and “trans" are italic in all cases.

In final words, the paper needs a minor revision.

Author Response

Dear Reviewer,

We are very grateful for the time you devoted to read our manuscript and for your helpful comments and suggestions. We did our best to make appropriate changes and correct the text so we hope our manuscript has improved. All suggested changes are highlighted (in yellow).

Below please find detailed responses to all queries.

Comment: In this paper, selected phenolic acids and flavonoids in various extracts f Glechoma hederacea prepared by different extraction methods owere analyzed by HPLC. In addition, antioxidant potential and cytotoxicity of the extracts were also determined.

Giving the plant Latin binomial names, the genus and species names in Latin should be in italic, which must be accompanied by their authority names, which should not be italic. Please indicate author name in the title, too.

Actually, please check all plant names and their families according to APG4 plant classification system as well as IPNI or the Plant List. Use the accepted names and new family names to be the most correct.

Plant family Latin names are not written in italic. Please correct.

Response: We corrected all plant names and their families in the manuscript to be written in italic. We also checked the plant names to be in line with the Plant List (http://www.theplantlist.org). We have also added author names to accompany the plant names as recommended. We did not make changes to the Latin plant names in References, even if there are mistakes, to keep the original title of the paper.

Comment: Grammatical and typological errors are present and should be rectified. For example;

  • line 17, “the cytotoxic activity of extracts” should be corrected as “cytotoxic activity of the extracts”.
  • line 164, correct as “IsoQ content in the water extracts”.
  • line 204, correact as “was determined by MTT assay”.
  • "per", "via", "versus", "e.g.", "i.e.", “viz.”  italic in all cases.
  • “p” (statistical) italic in all cases.

Response: We have implemented all recommended corrections.

Comment: line 30, author names for G. longituba, Rosmarinus officinalis, Plectranthus scutellarioides?

Response: We have added the authors' names accompanying the plant species names.

Comment: On which bases, did you select these particular phenolic acids and flavonoids in HPLC analyses? Why not more?

Response: We performed the HPLC analysis based on the standard phenolic acids and flavonoids we usually use in our laboratory. We used a number of phenolic acids (about 10) and flavonoids (about 8), but we were able to identify only these particular phenolic acids and flavonoids, which were in fact the predominant peaks in the chromatograms, while some minor peaks remained unidentified (i.e. we could not match the retention times to any of the standards we have).

Comment: Always use 3rd person style in the text. Not like “we decided to conduct …”

Response: Thank you for this suggestion, we have transformed the sentences into 3rd person style throughout the text.

Comment: line 332, please specify the plant species such as in “Hot water extract from the fresh Glechoma plant”

Response: We clarified which plant species was used to obtain the extract. We have added the relevant information to the manuscript.

Comment: line 344, which ethnobotanical use of the plant did you correlate the mentioned effect in “….wanted to verify whether the use of the plant may have a chemopreventive benefit….”?

Response: We wanted to examine the chemopreventive potential of the plant mainly in terms of its antioxidant properties. The protection from the excess of free radicals may be useful in many diseases, and in this case it could be correlated with arthritis, bronchitis, liver diseases, gastritis etc. We did not want to correlate the activity with one particular use of the plant, but rather indicate the additional protective role of the plant, widely used traditionally and ethnobotanically. We have modified the sentence.

Comment:

  • Please pay attention to punctuation marks.
  • line 388, correct as “phosphate-buffered
  • line 378, correct as “phenolic acids”.
  • post hoc should be italic in all cases.
  • Glechoma must be italic including references such as no. 1 and 10.
  • In “3rd”, rd must be superscripted.
  • In references, paper titles should be lower case except the first word in sentence.
  • line 579, correct as “Content of p-coumaric”, p should be italic in all cases.
  • 49, the terms "cis” and “trans" are italic in all cases.

Response: We have implemented all recommended corrections.